# Evaluation of VOCs Emitted from Biomass Combustion in a Small CHP Plant: Difference between Dry and Wet Poplar Woodchips

**DOI:** 10.3390/molecules27030955

**Published:** 2022-01-31

**Authors:** Enrico Paris, Monica Carnevale, Beatrice Vincenti, Adriano Palma, Ettore Guerriero, Domenico Borello, Francesco Gallucci

**Affiliations:** 1Council for Agricultural Research and Economics (CREA), Center of Engineering and Agro-Food Processing, Monterotondo (Roma), Via della Pascolare 16, 00015 Monterotondo, Italy; enrico.paris@crea.gov.it (E.P.); monica.carnevale@crea.gov.it (M.C.); beatrice.vincenti@crea.gov.it (B.V.); adriano.palma@crea.gov.it (A.P.); 2National Research Council of Italy, Institute of Atmospheric Pollution Research (CNR-IIA), Via Salaria km 29,300, 00015 Monterotondo, Italy; ettore.guerriero@iia.cnr.it; 3Department of Mechanical and Aerospace Engineering (DIMA), La Sapienza University of Rome, Via Eudossiana, 18, 00184 Rome, Italy; domenico.borello@uniroma1.it

**Keywords:** biomass combustion, POPs, CHP plant, VOCs, renewable energies, emission, organic pollutants

## Abstract

The combustion of biomass is a process that is increasingly used for the generation of heat and energy through different types of wood and agricultural waste. The emissions generated by the combustion of biomass include different kinds of macro- and micropollutants whose formation and concentration varies according to the physical and chemical characteristics of the biomass, the combustion conditions, the plants, and the operational parameters of the process. The aim of this work is to evaluate the effect of biomass moisture content on the formation of volatile organic compounds (VOCs) during the combustion process. Wet and dry poplar chips, with a moisture content of 43.30% and 15.00%, respectively, were used in a cogeneration plant based on a mobile grate furnace. Stack’s emissions were sampled through adsorbent tubes and subsequently analyzed by thermal desorption coupled with the GC/MS. The data obtained showed that, depending on the moisture content of the starting matrix, which inevitably influences the quality of combustion, there is significant variation in the production of VOCs.

## 1. Introduction

The use of renewable energy in the European Union has been estimated to increase to a share between 55% and 75% of total gross energy consumption by 2050. In the current energy situation, bioenergies represent the largest renewable and CO_2_-neutral energy source for the production of heat, electricity, and transport fuels [1]. Different biochemical or thermochemical conversion processes of biomass can be applied in order to produce power and heat, reduce the consumption of conventional fossil fuel sources, and represent a realistic threat to environmental sustainability [2].

The most appropriate biomass conversion process depends on biomass physical and chemical characteristics, such as moisture content, fixed carbon, volatile solids, C/N ratio, calorific value, ash and cellulose, hemicellulose, and lignin content, which are recognized worldwide as the main factors that affect the conversion processes’ efficiency [3].

Several studies have shown the feasibility of using residual biomass, such as shredded prunings or woodchips [4].

Biomass combustion represents the easiest way to generate energy from biomass [5], and it is currently defined as CO_2_-neutral because the carbon dioxide generated and emitted during the combustion phase is compensated by that which the biomass has absorbed during the growth phase [6].

In this context, biomass combustion in small combined heat and power plants (CHPs) represents a useful system for distributed renewable energy production, but, at the same time, close attention ought to be paid to the environmental aspects, with particular reference to the atmospheric pollution emitted from the biomass combustion process. The combustion conditions and the physical and chemical parameters influence the process and the related emissions [7,8,9,10,11,12].

In any combustion phenomenon, incomplete combustion products are generated. In particular, products of pyrolysis remain as unburnt generating particles and gases, such as carbon monoxide (CO), volatile organic compounds (VOCs), particulate matter (PM), and polycyclic aromatic hydrocarbons (PAHs), which contribute to environmental pollution and become a potential issue for the environment and human health.

From this point of view, it is important to evaluate the relationship between the biomass characteristics; the operating condition of the combustion plant and the flue gases produced in terms of particulate matter (PM_10_ and PM_2.5_); oxidized species, such as CO_2_, CO, SO_2_, and NO_x_; inorganic micropollutants, including heavy metals; persistent organic pollutants (POPs), including polycyclic aromatic hydrocarbons (PAHs); dioxins and furans (PCDD/Fs); and volatile organic compounds (VOCs) [13,14]. VOCs represent a very large amount of different organic compounds, whose emission strongly depends on the type of biomass used and the conditions of the combustion process [15]. These chemical compounds (CFCs, alkanes, alkenes, aldehydes, ketones, aromatic compounds, etc.) are defined by an “initial boiling point equal to or less than 250 °C measured at a standard pressure of 101.3 kPa”, as expressed in the European Directive on Air Quality 2008/50/CE.

VOCs can have different effects on humans and the environment depending on their chemical characteristics. They can be precursors of photochemical smog under sunlight radiation in the presence of nitrogen oxides (NO_x_) [16] (such as alkenes and alkanes from C3 to C8), they can have a high ozone depletion potential (such as CFCs and halons), they can be greenhouse gases (such as CFCs), or they can be directly toxic to humans (such as chlorinated compounds, benzene, etc.). In particular, it has been widely studied how some VOCs emitted by biomass combustion play an important role in tropospheric ozone and photo-oxidant production. For instance, during the smoke plume transport process, VOCs combined with NO_x_ can be oxidized to generate secondary organic aerosol (SOA) [17,18]. In the function of the described effects, Ciccioli et al. [19] proposed to classify the aforementioned classes as VOC-OX, VOC-STRAT, VOC-TOX, and VOC-CLIM, respectively, which are precursors of photochemical smog, harmful to the stratospheric ozone layer, toxic for humans, and climate-altering.

Several authors have previously investigated the emissions generated from domestic woodstoves and fireplaces of the same hardwood as European beech, Pyrenean oak, and black poplar, demonstrating that the production of such compounds depends on the type of combustion process, the type of plants, and the biomass characterization [20,21,22,23,24]. However, there are few works on the determination of VOCs emitted by Mediterranean vegetation species [19], and they are generally limited to the determination of PM and the main compounds in flue gases [23,25].

This work aims to investigate the emissions of VOCs by the combustion of woodchips from wet and dry poplar, which is one of the most important plantation trees in the European region [26,27]. Biomass moisture content is a parameter that is greatly affected by the type of biomass and by climatic and storage conditions, which negatively affects the combustion process, worsening the uniformity of the burning process. In this work, the VOCs generated by the poplar combustion process in the boiler were evaluated by modifying only the biomass moisture content. To understand the correlation between the physical and chemical biomass characteristics (lignin content, cellulose and hemicellulose content, ash content, carbon contents, hydrogen, nitrogen, sulfur, oxygen, heating value), combustion parameters, and emission of VOCs, a statistical evaluation was carried out.

## 2. Materials and Methods

### 2.1. Biomass Characterization

The biomass involved in combustion tests was poplar woody biomass harvested in the research area of CREA-IT in Monterotondo, Italy. The two poplar woodchips used were obtained from the same starting biomass but were differently treated: the first one (dry poplar, DP) underwent an open-air drying process, while the second (wet poplar, WP) was used directly after the harvesting. The biomass combustion test and related analysis were carried out by the Laboratory for Experimental Activities on Renewable Energy from Biomass (LASER-B) of CREA-IT of Monterotondo (Rome). The characterization concerned the determination of moisture content; higher heating value; ash content; and carbon, hydrogen, and nitrogen content. These determinations were made in triplicate. The moisture content, on a wet basis, was measured according to UNI EN ISO 18134-1:2015 and using a Memmert UFP800 drying oven. The higher heating value (HHV) was determined according to UNI EN ISO 18125:2018 and using a Paar 6400 isoperibol calorimeter, and the lower heating value (LHV) was calculated from the HHV and the hydrogen content. The total content of carbon (C), hydrogen (H), and nitrogen (N) was measured according to UNI EN ISO 16948:2015 and using a Costech ECS 4010 CHNS-O elemental analyzer. Ash content was measured according to UNI EN ISO 18122:2016 and using a Lenton EF11/8B muffle furnace.

The evaluation of heating value; ash content; and C, H, and N contents was carried out on a dry basis. In particular, with regard to the three analyses mentioned above, the dried sample obtained from the moisture determination procedure was ground first with the Retsch SM 100 cutting mill for a preliminary size reduction and then with the Retsch ZM 200 rotor mill. The lignin (Lign), cellulose (Cell), hemicellulose (Hem), and chlorine (Cl) content were obtained using the Phyllis database [28]. In order to obtain preliminary data about the thermal behavior of dry and wet poplar, the biomass was studied by thermogravimetric and differential scanning calorimetry analysis (TGA/DSC).

### 2.2. Experimental TGA Analysis

A thermoanalytical test is the most commonly used method to estimate the thermal kinetics of biomass during a thermochemical conversion process. TGA analysis provides data about the phase variation of the sample, mass loss, and emission production depending on the nature of the sample [29,30]. In this study, thermogravimetric curve and its derivative (DTG) allow us to investigate combustion process dynamics of dry and wet poplar by means of a Mettler Toledo TGA/DSC1 STAR^e^ System in the following operating conditions: temperature between 25 °C and 1000 °C; a heating rate of 80 °C/min; and an air flow rate of 60 mL/min.

### 2.3. Experimental Combustion Tests

The combustion tests were carried out through a demonstrative cogeneration plant based on a moving grate furnace (350 kW_th_) and equipped with a steam generator (500 kg/h at 1.2 MPa). The facility (Figure 1) was characterized by a cross-current combustion chamber and a secondary chamber for post-combustion. The biomass loading into the furnace occurred through a combined double auger loading system (DUPLO^®^) that allowed the use of even unconventional biomasses, and the mobile-grate system allowed the burning of biomasses with different particle sizes and with a moisture content up to 55% on a wet basis. The exhaust gases produced from biomass combustion were treated with a cyclone and a baghouse filter as PM abatement systems before the passage through the chimney. Along the chimney, several sampling points were established in order to evaluate the emissions in relation to the operating conditions of the furnace and the type of biomass.

For the PM sampling, a probe (HP5 Dadolab, Cinisello Balsamo MI, Italy) and an isokinetic sampler (ST5 Dadolab, Cinisello Balsamo MI, Italy) were used, respecting the European method [31]. PM was sampled by means of glass microfiber filters and quantified with gravimetric analysis using a Mettler Toledo AL104 Analytical Balance placed in a conditioned room at 20 °C and 50% humidity. The management of the biomass combustion process occurred by means of the control of parameters, such as biomass feeding, grate movement, and air distribution through blowers both inside the furnace and downstream of the filters.

### 2.4. Sampling and Analysis of VOCs

During the combustion test, the monitoring of VOCs was carried out according to UNI EN 13649, which represents the procedure for the characterization of single VOCs. In particular, such compounds were sampled by thermal desorption multilayer tubes through a dynamic dilution procedure using a “DDS-Tecora” system. This method is recommended when the concentration of water is high enough to cause the risk of condensation. According to this method, the flue gases are diluted with purified air in a mixing chamber before being sampled onto a thermal desorption tube. In this work, a 1:5 dilution factor and a flow of 50 mL/min were used for sampling. The sampling end of an identical, secondary back-up tube was connected to the outlet of the primary sampling tube as a check on breakthrough. Volatile organic compounds from poplar wood combustion were determined using a thermal desorption system (Markes TD 100 xr) coupled with a gas chromatographic mass spectrometer (Agilent 7000—7890A). The tubes were thermo-desorbed with a flow of 50 mL/min up to a temperature of 380 °C for 10 min with a 1:10 split. The analytes were collected on a focusing trap at the temperature of −10 °C and then desorbed for 1 min with a cold trap high temperature at 380 °C with a 40 °C/s rate. The capillary column used for the analysis was DB-502 (Agilent—60 m, 0.32 mm, 1.8 μm). After the determination, VOCs were classified into three groups: chlorofluorocarbons (CFC), chlorinated compounds (Chl), and aromatic compounds (Aro).

### 2.5. Statistical Analysis

In order to describe the biomass and the relative emission during combustion, principal component analysis (PCA) was performed through the PAST software (PAleontological STatistics) to investigate the correlation between physical and chemical parameters of biomass and the VOCs emission during the combustion process. PCA is a classical multivariate method widely used to interpret variation in a high-dimensional interrelated dataset with a large number of variables. It is a mathematical methodology that uses orthogonal transformation to convert a set of cases of possibly correlated variables into a set of values of uncorrelated variables which are known as principal components (PCs), thus reducing the number of variables. The two samples tested DP and WP were evaluated and compared through PCA, considering 12 variables: moisture; ash; carbon, hydrogen and nitrogen content (C, H, N); cellulose (Cell); hemicellulose (Hem); lignine (Lign); chlorofluorocarbons (CFC); chlorinated compounds (Chl); aromatic compounds; and chlorine (Cl). Since there were two materials studied, the only PC1 explained 100% of the total variability.

PCA loadings are the coefficients of the linear combination of the original variables from which the principal components (PCs) are constructed. In order to identify which variables have the greatest effect on sample variability, PC1 loadings were calculated. In this way, from the correlation between PC1 and the 12 variables, it was possible to identify correlations between the variables themselves.

## 3. Results and Discussion

### 3.1. Biomass Characterization

The chemical–physical characterization of biomass was carried out for the two different types of samples (dry and wet poplar), providing results on parameters that can affect the combustion quality and the related emissions. Biomass characterization results are reported in the following table (Table 1).

The results show comparable values in terms of elemental composition and calorific values content that meet the requirements for poplar chips used in the combustion process. On the contrary, moisture and ash content showed substantially different results and were mostly responsible for incomplete combustion and potentially harmful emissions. In the case of wet poplar, higher ash and moisture content results in a higher amount of unburnt compounds and bad combustion conditions. The sulfur content was below the limit of quantification (LOQ) and, hence, was considered negligible.

### 3.2. TGA Analysis

A preliminary analysis in TGA was conducted to evaluate the behavior of the matrices subjected to the thermal stress of combustion in a lab-scale test.

In Figure 2, the black curve (TGA) has two main steps that represent, respectively, up to about 100 °C due to the loss of water (the first step) and from about 200 °C up to about 720 °C due to the loss of volatile substances (the second step). The red curve (DTG) highlights these two stages with two peaks corresponding to the main thermal phenomena of mass loss.

In the graph of Figure 3, the black curve (TGA) has three distinct steps that represent the loss of moisture under 100 °C; the second step, which starts at 200 °C and finishes at about 350 °C, corresponds to the active zone and represents the hemicellulose and cellulose degradation. The last decomposition step, named the passive zone, from 350 °C to 750 °C, represents the slow lignin degradation, corresponding to a potential amount of VOC production as reported by the literature data [32]. The DTG curve reflects weight variation shown with the TGA highlight the mass loss in the steps corresponding to the composition of the sample. From the comparison of the curves in Figure 2 and Figure 3, it is observed that there are differences in the thermal behavior of the two samples due mainly to the higher moisture content in the wet poplar; consequently, it is more likely that the use of this matrix will lead to the formation of emission compounds from incomplete combustion. By comparing Figure 2 and Figure 3, it is evident that there is a strong difference in the thermal trend of the same biomass poplar sample with different moisture levels. In fact, dry poplar has a thermogravimetric curve whose course is much more linear than the curve of wet poplar. This indicates that wet poplar subjected to thermal stress will lead to a less homogeneous combustion phenomenon than dry biomass.

Figure 4 shows the comparison carried out by means of the DSC analysis related to Figure 2 and Figure 3. It can be observed that as the chamber temperature increases, there is a large exothermic peak of the matrices in the temperature range between 250 and 800 °C. By integrating these heat exchange curves, it can be seen that the dry poplar generates a heat of 163.29 J against the 121.36 J generated by the wet poplar. This result also confirms that the higher moisture negatively affects the combustion conditions.

### 3.3. VOCs in Emission

The assessment of VOCs in flue gases was carried out for two different monitoring campaigns related to two different poplar moisture conditions: dry and wet.

During these tests, the temperatures reached by the furnace bed (467 °C) and by the post-combustion chamber (793 °C) were monitored. Several different types of VOCs were found in terms of molecular weight (from freons to 1,2,4-trimethylbenzene) and chemical class (Table 2).

From the comparison of dry and wet poplar, it is possible to observe that the highest concentrations are obtained from wet poplar. Backup tubes were analyzed for the assessment of breakthrough volumes, and recovery values above 10% were not found for any analyte.

The analysis of the flue gases emitted by the combustion of two same-origin biomasses with different moisture content shows that the wet biomass produced a higher quantity of VOCs. This phenomenon is due to a higher moisture content that leads to a lower-quality combustion process. In fact, a higher quantity of VOCs is originated from an incomplete combustion phenomenon [33]. Once the VOC concentrations relative to two different moisture levels of the biomass were obtained, the authors thought of constructing the lines that can express an ideal linear trend in the increase in the concentrations of the VOC classes as a function of the increasing moisture content (Figure 5). It is interesting to note that the increase in CFCs and aromatic compounds follows the same type of trend, while the line relating to chlorinated compounds has a less marked increase. This is probably due to the fact that the formation of chlorinated compounds is essentially limited by the presence of chlorine in the matrix, which is the limiting reagent in the formation of the compounds. Moreover, as moisture increases, and, therefore, as combustion conditions worsen, it is likely that organic micropollutants that are heavier than VOCs are formed, such as PAH, PCB, and PCDD/F, whose individual molecules contain more chlorine atoms, thus reducing the formation of chlorinated VOCs. Although the graph is the ideal trend based on only two real points, it is interesting to note how the trend of VOCs formation is positive in all cases as a function of the increase in moisture. This ideal chart forms a basis for comparisons with other works that will be conducted under the same conditions. The authors conducted an in-depth bibliographic search to find other works on VOC emissions from biomass combustion on similar boilers in which the moisture degree of the incoming biomass was specified, but it was found that there is still a lack of studies on this subject.

### 3.4. Principal Component Analysis

The statistical evaluation shows that, in agreement with the bibliographic sources, a correlation can be established between biomass characteristics and VOCs production. Specifically, this study shows how the main variable represented by moisture, under constant burning conditions (type of plant, burning equipment, biomass, air supply, etc.), influences the amount of some classes of VOCs in emissions.

Some studies have evaluated the correlation between the emissions of some classes of volatile organic compounds from Mediterranean species, focusing on the relationship between the content of cellulose, hemicellulose, and lignin and the emission composition, showing a positive correlation between benzene and lignin [34,35,36,37,38,39]. Other studies show that hemicelluloses decompose during the thermal process and produce large amounts of volatiles at 200 °C [40]. It is well known that drying biomass fuel improves combustion efficiency, increases steam production, reduces air emissions, and improves boiler operation. Except for suspension-firing furnaces, wood-fired boilers and furnaces require a fuel moisture content of below 55% to 65% in order to sustain combustion. For wood-fired incineration, the optimal moisture content is generally much lower, between about 10% and 15% [41]. The water content influences the combustion and the volume of flue gas produced per energy unit. The heating value of the fuel decreases with increasing moisture content. For biofuels, which have a very high moisture content, some problems may occur during firing. High moisture content can cause ignition problems and reduce the combustion temperature, which in turn hinders the combustion of reaction products and consequently affects the quality of combustion [42]. Thermal property values, such as specific heat, thermal conductivity, and emissivity, vary with moisture content [43]. To better understand the relations between biomass moisture and combustion VOCs emissions, an analysis of flue gas and physical and chemical properties of biomass was conducted. The twelve variables coming from these analyses were used to perform a PCA analysis capable of describing the two samples WP and DP. The only principal component 1 (PC1) describes 100% of the sample’s variability.

In Figure 6, the PC1 loading for each variable can be observed. Large loadings (positive or negative) indicate that a particular variable has a strong relationship to PC1. The sign of a loading indicates whether a variable and a principal component are positively or negatively correlated. The variables that mostly influence PC1 behavior are CFC, Chl, aromatics, and moisture. These variables are all positively related to PC1 and, consequently, are positively correlated with each other. Positive loadings indicate that variables are positively correlated; an increase in one results in an increase in the other. So, higher moisture values of biomass correspond to higher CFC, Chl, and Aro emissions.

Since moisture content was the only parameter that differentiated between WP and DP, and since the combustion parameters were a set constant, it is evident that bad combustion due to biomass moisture content is responsible for the significantly higher production of VOCs.

## 4. Conclusions

The work proposed aims to highlight how the biomass combustion process is negatively affected by the moisture content of the biomass itself. Compared to other chemical-physical characteristics (e.g., content in ashes, metals present, etc.), moisture content can be reduced quite easily by intervening in the storage conditions or by drying the biomass in the open field under the sun, if seasonality and latitude allow it. The study showed that biomass moisture content leads to a general increase in all classes of VOCs emitted during combustion processes and, thus, inevitably affects the quality of the surrounding air where the phenomenon occurs. It should be noted that such concentrations are even lower than those that would be obtained in the case of uncontrolled combustion or in the phenomena of open burning (e.g., forest fires, domestic fireplaces, combustion of pruning on the sidelines, etc.). Several articles in the literature show the characterization of emissions in terms of emitted organic compounds, even if the majority [20,44] identify almost exclusively aromatic or semi-volatile compounds, neglecting freon and C short-chain chlorinated compounds. However, there are no works in which the formation of volatile organic pollutants is related to the moisture content of the biomass burned. PCA analysis was able to clearly compare two samples with single measurements of many variables and give a preliminary quantification of the relation between biomass moisture and VOCs production from combustion. In general, the work demonstrates how, in a real plant, the moisture of the starting biomass enormously influences the formation of all the classes of VOCs considered (aromatic, CFC, and chlorinated). This is because an increase in the moisture content of the matrix hinders the optimal combustion process and, therefore, leads to the formation of a greater concentration of incomplete combustion compounds (including VOCs).

## Figures and Tables

**Figure 1 molecules-27-00955-f001:**
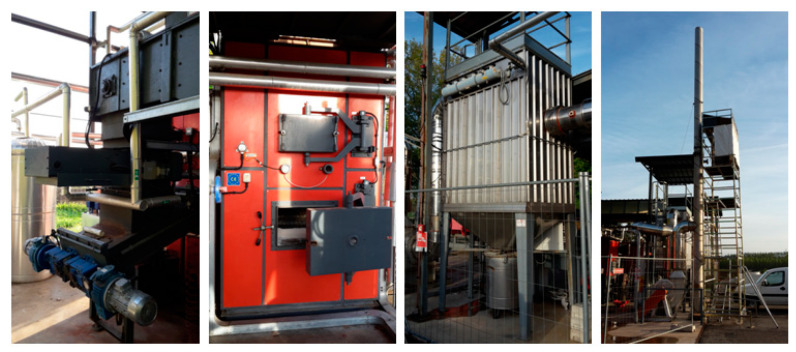
Details of the CHP plant (double screw mechanism, furnace, baghouse filter, and chimney).

**Figure 2 molecules-27-00955-f002:**
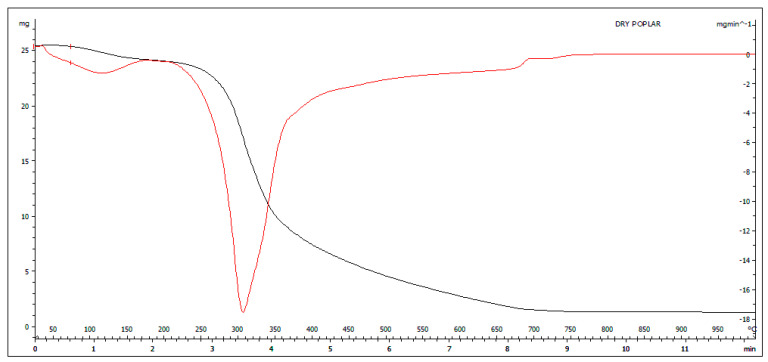
TGA (black curve) and DTG (red curve) of Dry poplar.

**Figure 3 molecules-27-00955-f003:**
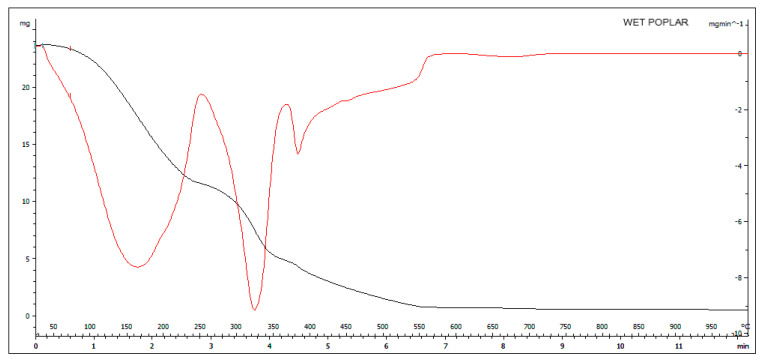
TGA (black curve) and DTG (red curve) of wet poplar.

**Figure 4 molecules-27-00955-f004:**
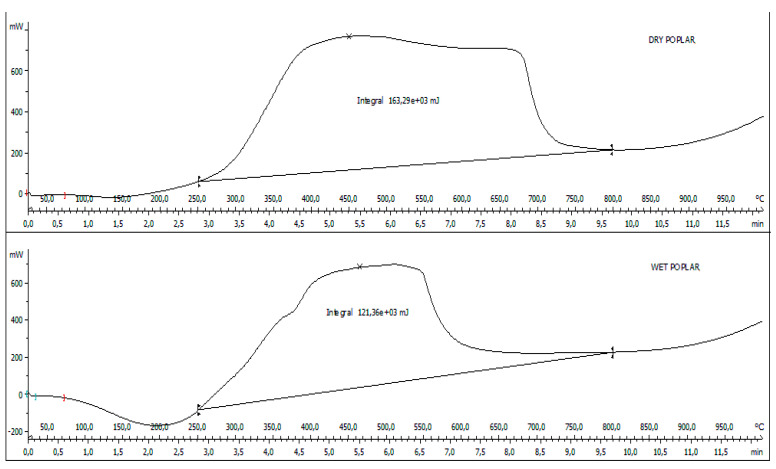
DSC curves, dry poplar, and wet poplar.

**Figure 5 molecules-27-00955-f005:**
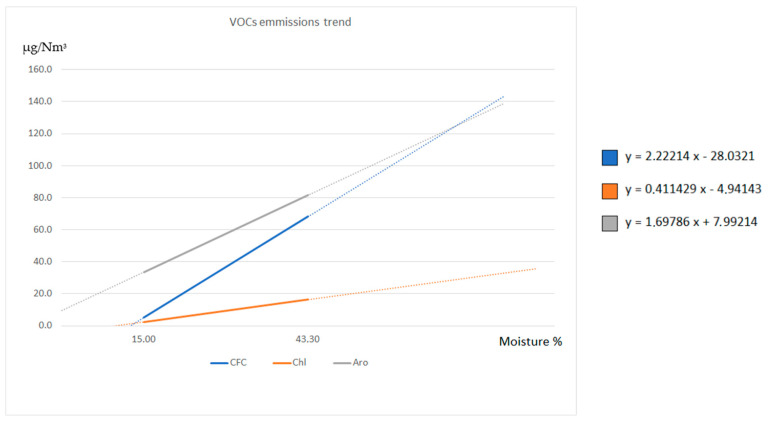
The ideal linear trend in the increase in the concentrations of the VOC classes as a function of the increasing moisture content.

**Figure 6 molecules-27-00955-f006:**
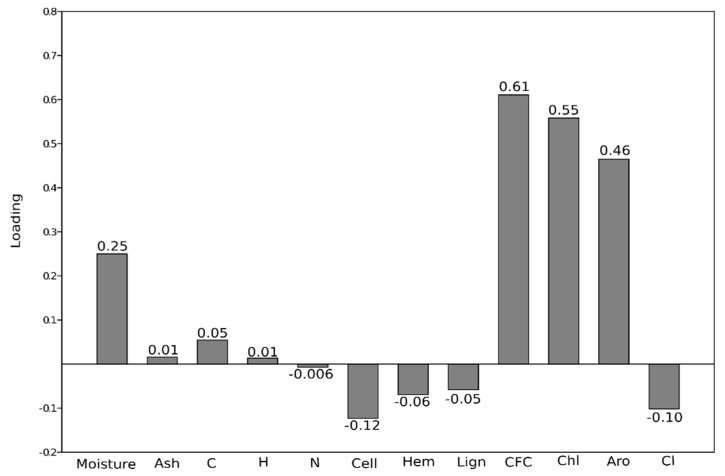
Loadings of PC1 relative to all the variables used for PCA analysis.

**Table 1 molecules-27-00955-t001:** Chemical and physical biomass parameters.

Parameters	Dry Poplar	Wet Poplar
Moisture %	15.00	43.30
Ash %	1.30	3.12
C %	41.20	47.36
H %	4.05	5.58
N %	1.10	0.31
HHV [MJ/kg]	18.27	17.94
LHV [MJ/kg]	17.43	16.77

**Table 2 molecules-27-00955-t002:** Concentration of VOCs in dry and wet poplar combustion emissions.

Compound	Dry Poplar (μg/Nm^3^)	Wet Poplar (μg/Nm^3^)
Dichlorodifluoromethane	0.10	0.38
Chloromethane	4.25	51.15
Bromomethane	<LOQ	14.55
Chloroethane	<LOQ	6.90
1,1-Dichloroethene	1.23	5.85
Dichloromethane	0.95	1.44
Benzene	30.97	60.51
Toluene	6.9	15.46
Chlorobenzene	1.33	5.32
p,m-Xylene	1.10	6.18
o-Xylene	0.45	2.83
Styrene	0.47	1.87
1,2,4-Trimethylbenzene	<LOQ	1.66

## Data Availability

Not applicable.

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
