# Peer review of "Evaluation of VOCs Emitted from Biomass Combustion in a Small CHP Plant: Difference between Dry and Wet Poplar Woodchips"

_molecules, 2022, doi:10.3390/molecules27030955_

Round 1
Reviewer 1 Report
Compared to the previous version, the article has been extended in terms of the description of the research methodology and the results obtained. Several remarks and suggestions for further improvement of the publication are indicated:
- the quality of figures 2, 3 and 4 should be improved;
- lines 265-267 “It is interesting to note that the increase in CFCs and chlorinated compounds follows the same type of trend, while the line relating to chlorinated compounds has a less marked increase” - there is an inaccuracy in the sentence;
- the discussion of the results obtained should be deepened.
Author Response
Compared to the previous version, the article has been extended in terms of the description of the research methodology and the results obtained. Several remarks and suggestions for further improvement of the publication are indicated:
- the quality of figures 2, 3 and 4 should be improved;
Answer 1:
Thanks for your suggestions.
We have improved the quality of the figures. In addition, the figures in higher resolution were sent to the journal in order to evaluate whether to improve further in the publication phase.
- lines 265-267 “It is interesting to note that the increase in CFCs and chlorinated compounds follows the same type of trend, while the line relating to chlorinated compounds has a less marked increase” - there is an inaccuracy in the sentence;
Answer 2:
Thanks for the comment. The sentence has been corrected as follows:
It is interesting to note that the increase in CFCs and Aromatics compounds follows the same type of trend, while the line relating to chlorinated compounds has a less marked increase.
- the discussion of the results obtained should be deepened.
Answer 3:
Thanks for the comment. We have deepened the results by expanding the discussion in paragraphs 3.2,3.3 and in the conclusions.

Reviewer 2 Report
This manuscript reports the effect of moisture content in biomass on the formation of volatile organic compounds (VOCs) during combustion. The manuscript is interesting in that the authors performed thermogravimetric analysis and GC/MS tests to detect the VOC content in the exhaust gas for wet and dry poplar wood with 43.30% and 15.00% moisture content, respectively, and performed principal component analysis (PCA) on the samples using PAST software (PAleontological STatistics). It was shown that there is a strong correlation between the water content of the biomass on the generated VOCs, especially the generation of Aromatic hydrocarbons and CFCs. This work has only done a relatively simple phenological analysis, with insufficient analysis of the raw data. There are two areas of confusing data interpretation throughout the paper, and more work is needed to elaborate the experimental results in more depth
Comment 1. There are several formatting errors in the article:
1) Misplaced chart layout, e.g. Table 1 is not aligned
2) Wrong use of superscripts, e.g. μg/Nm3 in Figure 4
Please check the format of the article carefully.
Comment 2. In Figure 2 and 3, the authors only perform a simple qualitative analysis and can only obtain the difference in thermal behaviour of the two samples due to the moisture content. There is no reliable data to support the conclusion that " it is more likely that the use of this matrix will lead to the formation of emission compounds from incomplete combustion". Please The authors further explain how this conclusion was reached by thermogravimetric analysis.
Comment 3. In Figure 4, the authors directly used the VOCS concentrations corresponding to the two samples at different water contents to construct the ideal equation. The straight line obtained by constructing the equation with only two sets of data may be subject to chance. It is suggested that the authors build the equation after adding several sets of data for samples with different water contents.
Comment 4. The use of "seems" in line 346 of the conclusion does not exactly indicate that the moisture content of the biomass plays a major role in the formation of CFCs and Aromatic compounds, which contradicts the description in the abstract. The author is requested to further implement the true conclusions of the article.
Comment 5. Some of the related work can be cited and discussed (Molecules 2017, 22(8), 1306; Nature Communications, 2022, 13, 295)
Author Response
Comment:
This manuscript reports the effect of moisture content in biomass on the formation of volatile organic compounds (VOCs) during combustion. The manuscript is interesting in that the authors performed thermogravimetric analysis and GC/MS tests to detect the VOC content in the exhaust gas for wet and dry poplar wood with 43.30% and 15.00% moisture content, respectively, and performed principal component analysis (PCA) on the samples using PAST software (PAleontological STatistics). It was shown that there is a strong correlation between the water content of the biomass on the generated VOCs, especially the generation of Aromatic hydrocarbons and CFCs. This work has only done a relatively simple phenological analysis, with insufficient analysis of the raw data. There are two areas of confusing data interpretation throughout the paper, and more work is needed to elaborate the experimental results in more depth.
Answer to Reviewer #2:
Dear Reviewer,
We thank you for your contribution to the revision of our paper. The manuscript reports the effect of moisture content in biomass on the formation of volatile organic compounds (VOCs) during combustion. We would like to specify that the analysis carried out in TGA is only a preliminary analysis to complete the characterization which provided us information on the thermal behavior of the different biomass investigated and which allows to simulate the dynamics of the combustion process as written in the first lines of the paragraphs 3.2.
The combustion tests, and VOCs sampling, were carried out by means of a real plant (350 kWth), as described in the text in paragraph 2.2. "Experimental combustion tests", as reported below:
“The combustion tests have been carried out through a demonstrative cogeneration plant based on a moving-grate furnace (350 kWth) and equipped with a steam generator (500 kg/h at 1.2 MPa). The facility (Figure 1) is characterized by a cross-current combustion chamber and a secondary chamber for post-combustion. […] The exhaust gases produced from biomass combustion are treated with a cyclone and a baghouse filter as abatement system before the passage through the chimney. Along the chimney several sampling points were established in order to evaluate the emissions in relation to the operating conditions of the furnace and the type of biomass […]”.
These combustion tests require large quantities of biomass to be tested, for this reason it is not easy and quite expensive to carry out many repeated tests.
Comment 1. There are several formatting errors in the article:
1) Misplaced chart layout, e.g. Table 1 is not aligned
2) Wrong use of superscripts, e.g. μg/Nm3 in Figure 4
Please check the format of the article carefully.
Answer 1:
Dear reviewer, thank you for the corrections. We have modified as per your instructions and conducted a format check of the paper
Comment 2. In Figure 2 and 3, the authors only perform a simple qualitative analysis and can only obtain the difference in thermal behaviour of the two samples due to the moisture content. There is no reliable data to support the conclusion that " it is more likely that the use of this matrix will lead to the formation of emission compounds from incomplete combustion". Please The authors further explain how this conclusion was reached by thermogravimetric analysis.
Answer 2:
Thanks for the comment.
We have expanded the description of the results of the TGA and we have also added the results related to the DSC (Differential Scanning Calorimetry). We have thus tried to better express the concept according to which, starting from the assumption that VOCs (like any organic micropollutant) originate in conditions of incomplete combustion, a dry biomass burns better than a wet one and consequently generates less voc.
Comment 3. In Figure 4, the authors directly used the VOCS concentrations corresponding to the two samples at different water contents to construct the ideal equation. The straight line obtained by constructing the equation with only two sets of data may be subject to chance. It is suggested that the authors build the equation after adding several sets of data for samples with different water contents.
Answer 3:
Thanks for the comment.
As reported in the text of the paper, the graph was ideally constructed on the basis of the tests conducted. Since the results obtained from the combustion of a boiler that consumes about 100kg / h of biomass, it was difficult to obtain other points. However, we are convinced that having used widely scientifically consolidated sampling and analysis methods worldwide and having conducted full-scale tests, leads to reliable results. The product chart has two functions:
- The first is to directly communicate the trends of increase in the formation of VOCs which are a function of the increase in humidity
- The second is to provide a basis for comparisons with other works that will be conducted under the same conditions.
It should be noted that the authors conducted an in-depth bibliographic search to find other works on VOC emissions from biomass combustion on boilers of similar power and in which the degree of humidity of the incoming biomass was specified, unfortunately the literature is poor.
Based on your observations, however, we have decided to correct the graph and reduce the ordinate axis so that it can be easier to interpret and to correct the text in this regard.
Comment 4. The use of "seems" in line 346 of the conclusion does not exactly indicate that the moisture content of the biomass plays a major role in the formation of CFCs and Aromatic compounds, which contradicts the description in the abstract. The author is requested to further implement the true conclusions of the article.
Answer 4:
Thanks for the tip, we have changed:
“In general, the work demonstrates how on a real plant, the humidity of the starting biomass enormously influences the formation of all the classes of VOCs considered (Aromatic, CFC and Chlorinated). This is because an increase in the humidity of the matrix hinders the optimal combustion process and therefore leads to the formation of a greater concentration of incomplete combustion compounds (including VOCs).”
Changes have also been made in the Abstract so that the text is consistent.
Comment 5. Some of the related work can be cited and discussed (Molecules 2017, 22(8), 1306; Nature Communications, 2022, 13, 295).
Answer 5:
Thank you for your suggestion.
We have read the suggested articles very carefully, however we did not find comparisons to be useful. There are several general notions about VOCs which, however, add nothing to what has already been written in the introduction.
However, we thank you for suggesting these articles that will surely be used in subsequent more appropriate works.
